# A Stackelberg-based repurchase strategy for rail freight options (BRFO)

Qi Shen[1], Tingyue Kuang[2], Jingwei Guo [2]*

**1** School of Business Administration, Faculty of Business Administration, Southwestern University of Finance and Economics, Chengdu, China, **2** Faculty of Business, City University of Macau, Macau, S.A.R. China

* 17828018959@163.com

## Abstract

This study presents a novel Buyback Rail Freight Option (BRFO), leveraging Stackelberg game theory to enhance the strategic management of rail freight transactions. By integrating traditional buyback theory with a multi-phase trigeminal tree pricing model and parameter identification through a nonparametric Ito stochastic method, the research addresses key challenges of information asymmetry and market uncertainty. The proposed methodology emphasizes dynamic pricing strategies and market adaptation, constructing a Nash equilibrium framework within railway freight pricing. The findings suggest significant strategic benefits for railway enterprises, positioning BRFO as a crucial tool for improving competitiveness in the face of alternative transport options.

## 1. Introduction

Railroad freight transport in the United States, characterized by its vast network of approximately 140,000 route miles, is one of the largest, safest, and most cost-efficient systems globally. This nearly $80 billion industry is primarily operated by private organizations, including Class I railroads and regional and local/short-line railroads, collectively providing over 167,000 jobs. The U.S. rail freight system stands out for its high investment in maintenance and capacity expansion, with railroad owners spending nearly $25 billion annually, representing about 19% of their revenues [1]. This system plays a crucial role in moving bulk commodities such as agriculture, energy products, and intermoal traffic, offering significant benefits over other transportation modes, such as reduced road congestion, highway fatalities, fuel consumption, greenhouse gas emissions, logistics costs, and public infrastructure maintenance costs [2].

The potential for electrification of the freight rail network has been a topic of discussion, particularly in the context of reducing greenhouse gas emissions and addressing climate change. However, the U.S. freight rail industry, including railroads and equipment manufacturers, views electrification as a less viable option due to the high costs involved, estimated to be as high as $4.8 million per track mile [3]. Instead, the industry is exploring alternative fuel sources like biofuels, hydrogen fuel cells, and zero-emissions battery cells to power freight locomotives. These alternatives are considered more feasible and cost-effective in reducing

**Data Availability Statement:** All relevant data are within the manuscript.

**Funding:** This work was supported by the National Natural Science Foundation of China (No. 61803147). The funders had no role in study

design, data collection and analysis, decision to publish, or preparation of the manuscript.

**Competing interests:** The authors have declared that no competing interests exist.

emissions without the need for extensive infrastructure changes that electrification would necessitate [4].

Amidst this backdrop, the prevailing issue of opaque market information necessitates innovative approaches in freight pricing and contractual frameworks to align with the evolving market and customer demands. Herein lies the significance of game theory and signaling-whereby introducing options and repurchase agreements serve as strategic signals indicating service quality and reliability, akin to a commitment by rail companies towards their offerings [5]. The significance of this study lies in its novel approach to tackling the inherent uncertainties and risks in rail freight transactions through the introduction of the Buyback Rail Freight Option (BRFO). This model integrates Stackelberg game theory, offering a structured framework that addresses information asymmetry and market volatility. One of the major challenges is the accurate modeling of market dynamics and customer behavior, which requires sophisticated mathematical tools and comprehensive data. The advantages of the BRFO model include enhanced strategic management, dynamic pricing flexibility, and improved market responsiveness. However, there are also drawbacks, such as the complexity of implementation and the need for precise parameter estimation to ensure the model's effectiveness. Despite these challenges, the potential benefits of adopting the BRFO model in rail freight operations are significant, positioning it as a crucial tool for improving competitiveness in the face of alternative transport options.

Leveraging the Stackelberg game theory framework, we establish a structured approach for pricing and risk management within the rail freight sector [6]. This model delineates a leader-follower dynamic, typical of Stackelberg scenarios, where railway companies set prices and conditions that customers then respond to. This hierarchical decision-making process accommodates the fluctuating demands of freight and the variable operational costs, offering a methodical way to navigate market uncertainties [7]. Such a strategic configuration not only increases market transparency and mutual trust but also promotes fairer and more effective outcomes in the freight pricing domain.

The adoption of buyback mechanisms aligns seamlessly with the principles of Stackelberg game theory, where railway companies, acting as leaders, can set strategic parameters that customers, as followers [8]. Buyback policies, by offering the flexibility to repurchase contracts, enhance the railway industry's ability to respond to market volatility and customer needs swiftly. This strategic move not only stabilizes the market but also promotes a more agile and resilient rail freight sector Furthermore, the establishment of a nash equilibrium through buyback initiatives ensures a harmonious balance between the expectations of the market and the operational realities of the rail freight industry [9]. This equilibrium fosters an environment conducive to growth, attracting new investments and broadening the customer base, thereby underscoring the practicality and effectiveness of incorporating buyback strategies within the Stackelberg model framework.

While the Surface Transportation Board plays a crucial role in ensuring rate fairness within the rail freight industry, there's a growing consensus on the necessity for innovative pricing and contractual frameworks to align with evolving market dynamics and customer preferences [10]. This necessitates the exploration of advanced mechanisms like freight options and buybacks, providing enhanced flexibility and sophisticated risk management solutions for both service providers and their clientele.

The incorporation of these financial instruments has the potential to inject vitality into the rail freight sector [5]. By safeguarding against fluctuations in pricing and enabling more agile pricing models, these tools empower rail freight operators to swiftly respond to changes in the market landscape, foster enduring relationships with customers through long-term agreements, and efficaciously manage associated risks. Such forward-thinking approaches are

poised to not only draw in new clientele and investment but also significantly contribute to the sector's enduring growth and environmental sustainability [11].

Since the Chinese government's initiation of rail freight reforms in 2014, aimed at boosting competitiveness and market orientation, significant strides have been made towards enhancing the efficiency of railroad freight transportation. However, the sector continues to grapple with outdated pricing structures, particularly in an era where fixed pricing proves inadequate for the volatile market landscape [12]. The persistence of such antiquated pricing methods has been a bottleneck for the sector's growth, as evidenced by the modest annual increase in rail freight, trailing behind road freight advancements.

In response to these challenges, the scholarly community has delved into various innovative pricing strategies, aiming to align sales with optimal customer segments and timing to maximize revenue. Despite the promise of these differentiated pricing models, their practical application within the bulk freight market remains constrained by the complexities of customer segmentation [13]. This limitation, coupled with the prevalent practice among rail freight operators to secure contracts with key clients, underscores a critical area for reform. The current contract-based approach, while beneficial in retaining business, raises questions regarding its compatibility with dynamic market conditions and the principles of Stackelberg game theory [14].

Thus, reorienting the rail freight pricing strategy through the lens of Stackelberg game theory not only addresses the existing gaps in pricing models but also propels the sector towards a more sustainable, competitive future. This strategic pivot could pave the way for more equitable pricing strategies, enhancing the sector's responsiveness to market changes and fostering a healthier, more robust rail freight market in China. Incorporating Stackelberg principles into contract pricing models such as QF contracts, concurrent contracts [13, 15], and revenue-sharing contracts [16] could offer a more robust framework for handling the uncertainties and asymmetric information in freight pricing. Buyback approaches, which emphasize probabilistic reasoning and updating beliefs with new evidence, could enhance the design of freight options, making them more adaptable to market dynamics and risk factors. This integration could significantly improve the strategic decision-making process in railway freight operations, making it more aligned with market competitiveness and risk management strategies [17].

The implementation of buyback mechanisms within the rail freight industry can be effectively framed within the Stackelberg leadership model, a strategic fit that underscores the hierarchical decision-making process characteristic of this framework [18]. In the Stackelberg setting, railway companies, acting as market leaders, can proactively set buyback policies that preemptively address potential defaults and market fluctuations. This preemptive strategy allows these companies to dictate the terms of engagement in a way that followers—customers in this context—can adapt their decisions based on the established policies [19].

Consequently, considering the financial technology can indeed spread risk and deal with asymmetric information, several investigations in transportation economics attempt to develop freight options to protect against the risk brought on by changes in freight rates [1]. This approach, aligning with nash equilibrium, enables more dynamic and informed decision-making, adapting to market fluctuations and enhancing competitiveness. Particularly, it has been shown that Asian-style options, fuzzy options, power-Asian options, and index Asia options play an important role in mitigating risks in the shipping market. The application of financial technology to rail freight, particularly through freight options like Asian-style and power-Asian options, reflects a move towards using advanced financial mechanisms to manage risk and asymmetric information in transportation [20–22]. Nevertheless, all of them are not designed for trading. Unlike maritime transport companies, railway transport companies

are more concerned with expanding the railway transportation's competitiveness in the market and strengthening the existing condition of railway freight operations with the assistance of freight options. To address the need, a tradable rail freight option based on the features of rail freight transport has been proposed.

Applying Stackelberg game theory to tradable rail freight options offers a strategic lens through which to address market dynamics and manage default risks. This approach, drawing on the hierarchical decision-making model of Stackelberg dynamics, aligns with the broader applications of the theory across various sectors. For foundational insights into Stackelberg analysis and its implications for buyback strategies provide a comprehensive exploration, emphasizing the strategic utility of buyback approaches in navigating decision-making complexities [23].

In the practical realm of rail freight, delinquencies pose an inevitable challenge. When defaults occur, railway companies are compelled to swiftly reallocate the pre-booked capacities, often resorting to the spot market. This urgency is compounded by the limited timeframe within which to secure new clients, exacerbating potential losses for rail transport providers [20]. The inclusion of default clauses in rail freight options becomes crucial to mitigate such risks. However, the traditional default trigger mechanisms often present a dichotomy, reflecting a marked discrepancy in the perceived default conditions between railway companies and their customers, with the latter generally facing a higher default likelihood.

To address this imbalance and the paucity of research on rail freight option pricing that incorporates buyback strategies, this paper proposes a novel framework that integrates buyback policies as a strategic response to customer defaults. This initiative not only aims to bridge the gap in existing literature but also to delineate a clear pathway for the practical application of Stackelberg principles in crafting more resilient and adaptable rail freight pricing models. The subsequent sections of this paper will delve into the intricate details of this proposed framework, focusing on key issues such as the design of the transaction process for the buyback rail freight option and the pricing mechanisms that underpin this innovative approach. To construct the theoretical framework, this paper will concentrate on the following issues:

1. How will the transaction process of the buyback rail freight option be described?

2. How will the buyback rail freight option be priced?

In addressing the challenges presented by rail freight pricing and contract models, we introduce an innovative solution: the Buyback Rail Freight Option (BRFO). This new option is designed not only to enhance the adaptability of rail freight operations to market changes but also to catalyze the sector's growth. The development of a specific pricing model for the BRFO aims to identify the optimal pricing strategy, ensuring that rail freight transit becomes more attractive and viable. The key contributions of this paper are structured as follows:

Introduction of the BRFO: We present a novel BRFO, articulating its transactional framework within a Stackelberg model. This approach allows us to dynamically update the transaction process in response to market fluctuations and the various uncertainties characteristic of freight logistics.

Development of a Composite Algorithm: To effectively manage the intricacies of BRFO contracts, we propose a sophisticated algorithm that merges multi-phase trigeminal and Stackelberg game theories. This algorithm incorporates the principles of buyback strategies, enabling a probabilistic approach to decision-making that leverages both historical data and emerging market information to formulate optimal contract strategies.

Estimation of Key Parameters: Our model's accuracy hinges on the precise estimation of critical parameters. To this end, we utilize a nonparametric Ito stochastic method, enhanced by the strategic insights offered by Stackelberg theory. This enhancement allows for a more nuanced assimilation of past data and expert knowledge, culminating in a resilient and flexible pricing framework for the BRFO.

The remainder of this paper is organized to systematically explore and expand upon these contributions:

Section 2: Literature Review and Theoretical Background

This section delves into the realm of railway freight pricing research, juxtaposing existing literature with our novel approach that infuses Stackelberg game theory into the analysis. Prevailing studies in this domain explore a range of market-based solutions to the intricacies of railway freight pricing, especially in the wake of market deregulation. These investigations underline the potential of deregulation to bolster service provider efficiencies and spawn innovative pricing strategies.

Section 3: Methodology and Application

Here, we detail our methodological approach, outlining the application of Stackelberg strategies to the transactional dynamics, pricing, and strategic decision-making processes inherent to the BRFO. This comprehensive overview provides a clear roadmap for our analytical methods.

Section 4: Empirical Analysis and Results

We present the results of our empirical analysis, demonstrating the effectiveness of the BRFO model in enhancing strategic management, dynamic pricing flexibility, and market responsiveness in rail freight operations.

Section5: Discussion and Conclusion

Through this structured approach, we aim to offer a fresh perspective on rail freight pricing, informed by the strategic depth of Stackelberg game theory. We discuss the implications of our findings and present a viable pathway towards more dynamic and responsive freight logistics solutions.

## 2. Materials

This section delves into the realm of railway freight pricing research, juxtaposing existing literature with our novel approach that infuses Stackelberg game theory into the analysis.

Prevailing studies in this domain explore a range of market-based solutions to the intricacies of railway freight pricing, especially in the wake of market deregulation [24–26]. These investigations underline the potential of deregulation to bolster service provider efficiencies and spawn innovative pricing strategies. However, the promise of deregulation is often curtailed by the presence of competitive monopolies, propelling the need for sophisticated financial instruments, akin to those employed in freight markets, to circumvent such challenges. The integration of option contracts into the freight pricing equation, exemplifies the utility of these financial strategies in synchronizing supply chain activities, thereby augmenting carrier profits and diminishing shipper expenditures [27, 28].

Building on this foundation, our research introduces a Stackelberg framework to the discourse, emphasizing the potential of employing probabilistic methods and leveraging the Stackelberg equilibrium to refine freight pricing strategies [29, 30]. By adopting this strategic viewpoint, our study aims to transcend the limitations posed by deregulation and competitive market dynamics, fostering a pricing model that is not only more responsive to market shifts but also grounded in a rigorous and informed decision-making process [31]. This enhanced

approach aspires to elevate the overall efficiency and strategic alignment within the railway freight sector.

Before 2012, the exploration into the utility of options contracts within railway freight pricing was nascent [24, 30]. The incorporation of such contracts was posited to significantly enhance the efficiency of the railway freight market. Following this, analytical models like binary and trinomial trees emerged to fine-tune the benefits for both service providers and their clientele. However, these studies mainly concentrated on determining the ideal quantity of rail freight options to be purchased, overlooking the need for a mechanism for railways to repurchase these options, a crucial step for broadening risk management strategies.

To bridge this gap, the concept of buyback, prevalent in stock markets, was considered for its potential applicability in the railway freight sector. This strategy involves the repurchase of unsold freight or goods post the final transaction, a practice scrutinized in various studies [29]. They notably identified that buyback contracts could potentially elevate customer purchase intent beyond the capability of other financial instruments, thereby serving diverse market scenarios and enhancing the overall risk management and revenue generation of the supply chain [9].

Further exploration into buyback strategies across different markets revealed their utility in optimizing supply chain efficiency [5] and addressing complex network challenges within smart power supply chains. However, these strategies necessitate substantial collaboration among supply chain stakeholders, raising concerns about trust and the balance of benefits. In addition, the broader application of game theory in buyback strategies encompasses various competitive game models such as Nash games, differential games, and fuzzy games [32–34]. For instance, Buyback contracts have been extensively studied within the context of competitive environments. Hu et al. [32] examined a two-stage supply chain composed of a risk-neutral supplier and multiple competing retailers. They investigated the impact of competition and retailers' loss aversion on decision-making behavior and coordination of the supply chain with a buyback contract. Moreover, Zhao [34] focused on the coordination of supply chain systems under both price and inventory competition. This study explored how a common supplier could effectively coordinate the downstream retailers engaged in competitive pricing and inventory management through the implementation of buyback contracts.

Given the sporadic application of buyback mechanisms in railway freight transactions, which have traditionally relied on less dynamic transactional standards, there's a clear need for ongoing research in this area. This paper advocates for the integration of buyback and options contracts within the railway freight framework, proposing a novel theoretical model that combines traditional buyback theory with advanced multi-phase trigeminal tree pricing. This model aims to leverage the strategic benefits of buyback mechanisms, enhancing the adaptability and strategic depth of railway freight pricing strategies.

## 3. Methods

### 3.1 Definition of BRFO

In the realm of rail freight logistics, akin to tangible commodities, services necessitate contractual agreements. However, freight services differ fundamentally from physical commodities as they represent a commitment to transport goods rather than the goods themselves. To address this unique aspect and mitigate associated risks, we introduce the concept of the Buyback Rail Freight Option (BRFO).

*Definition 1* The BRFO is conceptualized as a contractual agreement granting customers (holders) the flexibility to adjust to fluctuating freight rates over a specified period, culminating

in the execution or expiry of the contract. Concurrently, rail transport companies (writers) assume the obligation to offer a buyback option under predefined market conditions.

In essence, the BRFO serves as a financial instrument tied to the service of rail freight transportation. It empowers rail companies with the strategic choice, though not the obligation, to initiate buyback actions based on prevailing market dynamics. This capacity to repurchase options is particularly critical when confronting scenarios of limited freight capacity or significant rate fluctuations in the spot market. By exercising the buyback option, rail companies can effectively manage risk and ensure the strategic deployment of freight services, thereby maintaining operational fluidity and market responsiveness.

## 3.2 Transaction process description

To comprehensively examine the pricing strategies encompassing Options and Buyback methods, we dissect the pricing process into two distinct phases: the Contract Market Period (T = 0) and the Market Period (T = 1). This delineation aims to illuminate the strategic decisions undertaken by both the railway freight enterprise and the contracting customer at each juncture.

Accordingly, this pricing dynamic can be conceptualized as a two-stage Stackelberg game, where the railway freight company assumes the role of the leader, setting the terms and conditions of the freight contracts, and the customers, as followers, make their decisions based on the established framework. This hierarchical interaction underscores the sequential decision-making process inherent in the Stackelberg model, emphasizing the proactive stance of the railway companies in shaping the market landscape. The sequence of actions within this game-like structure unfolds as follows:

Step 1. In the Contract market ($T = 0$), the railway freight enterprise implements how much prices in option contracts ($w_0$) and option execution ($w_1$) will be sold to the contract customers to forward the market demand and make some preparations (short or long period) for the freight transportation by the train departure (after $T = 1$). The railway freight enterprise also makes the Buyback price ($w_b$) and percentage ($M$) simultaneously. The contract customers must ensure the purchasing amounts ($N$) and pay the option fees ($w_0N$) to the railway freight enterprise.

Step 2. According to the published price strategy of BRFO, the contract customer decides the purchase amount of BRFO to maximize the expected returns.

Step 3. In the Spot/repurchasing market, the railway freight enterprise will decide whether or not to repurchase the sold options based on the spot market freight rate ($s_t$) and the option strike price ($w_1$).

Step 4. The contract customer will obtain the buyback fees if the railway freight enterprise trigger repurchase. Next, they will repurchase the required capacity in the spot market.

In conclusion, the game sequence is demonstrated in Fig 1.

This flowchart visually represents the methodology, highlighting the step-by-step process of implementing the BRFO model, including the decision-making points and interactions between the railway freight enterprise and the contract customers.

## 3.3 Symbol and assumption

For clarity and precision in our analysis, we've compiled a comprehensive list of symbols utilized throughout this study, which is systematically organized in Table 1. Each notation, while

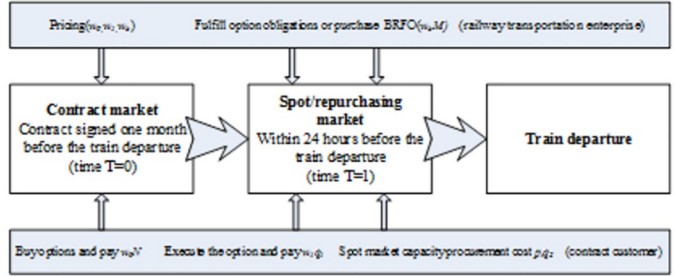

**Fig 1. The Transaction process of BRFO.**

briefly introduced here, will be expounded upon with greater detail in subsequent sections as they become contextually relevant to the discussion.

Furthermore, in order to build the mathematical model, the following assumptions are used in this work.

**Assumption 1**. This article's Advanced option is the European Selection, which can only be exercised on the expiration date.

**Assumption 2**. A supply chain is considered as two participants (railway freight service providers and contract customers). Both of them are significantly rational and risk aversion. The "Optimal decision" for all sides means the objective of maximizing supply chain's utility.

**Table 1. Summary of notations.**

| Notation | descriptions |
|---|---|
| $w_0$ | option price(unit capacity) |
| $w_1$ | option strike price(unit capacity) |
| $w_b$ | buyback price(unit capacity) |
| $s_t$ | spot market freight rate (unit capacity) in time t |
| $p_u$ | *probability of $s_0$ increase* |
| $p_d$ | *probability of $s_0$ drop* |
| $p_m$ | probability of $s_0$ remain unchanged |
| $w_u$ | value of RFO at expiration when $s_t$ increased |
| $w_d$ | value of RFO at expiration when $s_t$ drop |
| $u$ | *the advance ratio of $s_0$* |
| $d$ | *the decline ratio of $s_0$* |
| N | the purchase amount of RFO |
| M | the buyback percentage |
| Q | the sunk cost |
| c | fixed cost per unit capacity |
| $q_1$ | the RFO's execution amount |
| $q_2$ | the amount that the capacity buys on the spot market |
| $b_1$ | extended planning transportation expenses (per unit capacity) |
| $b_2$ | short-term transportation preparation costs (per unit capacity) |
| K | The overall capacity offered by rail freight transportation |
| $\alpha$ | standard deviation of railway freight market returns |
| D | market capacity consumption |
| f(D) | probability density function of freight demand |
| r | risk-free rates of interest |

**Assumption 3**. In this model, the freight rate in the railway freight charge market is an unusual externality that is totally determined by the external market's economic circumstances and is unaffected by any supply chain actors.

**Assumption 4**. The buyback and spot markets follow a typical distribution of freight demand, respectively:

$$f(\varepsilon) = 1 \Big/ b - a (a < \lambda < b); f(\lambda) = 1 \Big/ b - c (c < \lambda < d).$$

**Assumption 5**. The buyback method is partly buyback. The railway freight service providers will partly repurchase the remaining option contracts as the applying standard after the option expiration's date. The buyback price is fixed.

## 4. Results

According to the transaction procedure, the repurchasing market has two distinct probable status: Repurchase all option contract or not.

### 4.1 The expected profit from all repurchasing options

If every option contract is repurchased, it is shown that they cannot be executed. This happens on the premise that the spot market prices $s_t$ is higher than the option strike prices $w_1$. For the contract customers, they will lose the "opportunity sunk cost" due to they have to buy freight service on the spot market. In detail, the contract customers initially appreciate accomplishing all charges in the contract market. If there are remaining charge transactions within the spot market period, contract customers will lose their "initial appreciation". Because if they do not plan to purchase on the contract market from the onset, they may immediately charge on the spot market without preparing works, which requires money and focus ("opportunity sunk cost"). According to Ronayne et al. (2021), "opportunity sunk cost" may be assessed and converted to other aspects of economic activity value. To compute the supply chain's utility, suppose suck cost in every unit of the charge options in the spot market are related to the option execution prices $w_1$. Sunk cost "$Q$" might be described as the following:

$$Q = (Lw_1)^t \quad (0 \le L \le 1) \tag{1}$$

Although this suck cost will arise with the time consuming, suppose the T = 1 in whole spot market charging period to calculate conveniently. Thus, it is simple to measure the expected profit function from repurchasing options as follows:

$$\xi_1 = pD + (\omega_b M - \omega_0)N - (\omega_d + Q)D \tag{2}$$

$$\zeta_1 = (w_d - b_2)D + w_0 N(1 + r)^t - (w_b M + b_2)N - KC \tag{3}$$

where $\xi_1$ represents the utility of contract customer, $\zeta_1$ describes the utility of railway transportation enterprise from repurchasing all options.

Additionally, the average utility function of the supply chain is provided as follows:

$$E(\xi_1) = \int_{-\infty}^{\infty} [(p - \omega_d - Q)D + (\omega_b M - \omega_0)N]f(x)dx \tag{4}$$

$$E(\zeta_1) = \int_{-\infty}^{\infty} [(w_d - b_2)D + (w_0(1 + r)^t - w_b M - b_2)N - KC]f(x)dx \tag{5}$$

## 4.2 The expected profit from partly repurchasing options

The initiation of buyback behavior is contingent on the market environment, while the amount of repurchase is set by the railway transport enterprise. They elect to repurchase all or a portion of options depending on predicted market risks and expected profits. Thus, we cannot exclude an alternative which is to partial repurchase options. The buyback standard could not be attached if the quantity of freight service (D) purchased by the contract customer exceeds the amount of existing option contracts (N). The following describes the expected profit function from partial repurchase options:

$$\xi_2 = \begin{cases} pD - w_0 N - w_1 D + w_b M(N - D), D \leq N \\ pD - (w_0 + w_1)N - (w_u + Q)(D - N), D > N \end{cases} \tag{6}$$

$$\zeta_2 = \begin{cases} (w_1 - b_1)D + w_0 N(1 + r)^t - KC - ((w_b + b_2)(N - D)M), D \leq N \\ (w_1 - b_1)N + w_0 N(1 + r)^t + (w_u - b_2)(D - N) - KC, D > N \end{cases} \tag{7}$$

where $\xi_2$ represents the utility of contract customer, $\zeta_2$ describes the utility of railway transportation enterprise from partially repurchasing options.

Moreover, the average utility function of the whole supply chain is respectively delivered as follows:

$$E(\xi_2) = \int_{-\infty}^{N-a+bp} [(p - w_1)D - w_0 N + w_b M(N - D)]f(x)dx$$
$$+ \int_{N-a+bp}^{\infty} [pD - (w_0 + w_1)N - (w_u + Q)(D - N)]f(x)dx \tag{8}$$

$$E(\zeta_2) = \int_{-\infty}^{N-a+bp} [(w_1 - b_1)D + w_0 N(1 + r)^t - KC - ((w_b + b_2)(N - D)M)]f(x)dx$$
$$+ \int_{N-a+bp}^{\infty} [(w_1 - b_1)N + w_0 N(1 + r)^t + (w_u - b_2)(D - N) - KC]f(x)dx \tag{9}$$

## 4.3 Parameter solution

The determination of parameters is considered as a crucial aspect of the model-solving process. This study proposed a nonparametric Ito stochastic approach for the estimation of the key parameters. The extension of the Binomial Tree yields a Trinomial Tree, as is well knowledge. The basic Trinomial Tree is shown in Fig 2.

As the number of periods increases, the Trinomial Tree model faces the issue of determining the parameters including the magnitude and probability of $s_0$ growth and decrease in the spot market. Therefore, it is necessary to adjust the parameters to maintain the same standard deviation of railway freight market returns. Initially, discretize the continuously changing underlying prices for assets ($s_t$).

Dividing time $[0,T]$ into $n$ equal parts, $s_t$ is subject to geometric Brownian motion within time $[t_i, t_{i+\Delta t}]$. This equation can be defined by

$$\frac{\Delta S}{S} = \alpha \Delta t + \sigma(B(t_{i+\Delta}) - B(t_i)) \tag{10}$$

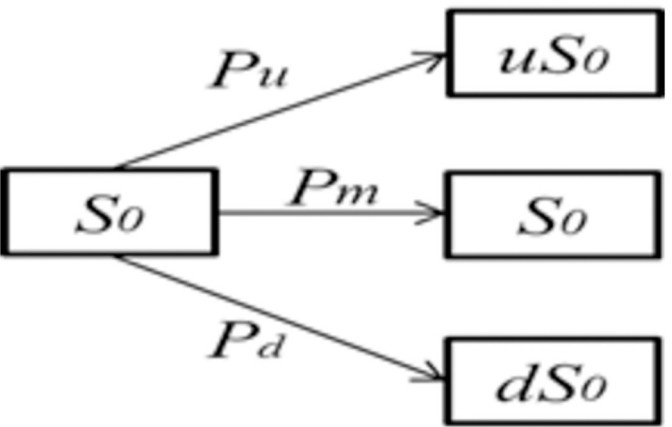

**Fig 2. The basic Trinomial Tree.**

The Ito stochastic integral may be used to update the formula (15) as follows:

$$S(t) = S(t_i)e^{\sigma(B(t_{i+\Delta})-B(t_i))+(\alpha-\frac{\sigma^2}{2})(t-t_i)} \tag{11}$$

where $e^{\sigma B(t)-0.5\sigma^2}$ is the equivalence of martingale measure. Use the fact that $e^{\sigma B(t)-0.5\sigma^2}$ is a martingale and $E[e^{\sigma B(t)-0.5\sigma^2}] = 1$ to conclude that:

$$E[S(t)] = E[S(t_i)e^{\sigma(B(t_{i+\Delta})-B(t_i))+(\alpha-\frac{\sigma^2}{2})(t-t_i)}] = S(t_i)E[e^{\sigma(B(t_{i+\Delta})-B(t_i))-\frac{\sigma^2}{2}(t-t_i)}e^{\alpha(t-t_i)}] = S(t_i)e^{\alpha(t-t_i)} \tag{12}$$

Furthermore, as per the Ito lemma, the function of an Ito process remains an Ito process; hence, the nonparametric Ito stochastic technique may be utilized to produce as:

$$S^2(t) = S^2(t_i)e^{2\sigma(B(t_{i+\Delta})-B(t_i))+(2\alpha-\sigma^2(t-t_i))} \tag{13}$$

$$S^3(t) = S^3(t_i)e^{3\sigma(B(t_{i+\Delta})-B(t_i))+(3\alpha-\frac{3\sigma^2}{2})(t-t_i)} \tag{14}$$

where $e^{2\sigma B(t)-2\sigma^2}$ and $e^{3\sigma B(t)-4.5\sigma^2}$ are the equivalence of martingale measure. Here, use the fact that $E[e^{2\sigma B(t)-2\sigma^2}] = 1$ and $E[e^{3\sigma B(t)-4.5\sigma^2}] = 1$ to conclude that:

$$
\begin{aligned}
E[S^2(t)] &= E[S^2(t_i)e^{2\sigma(B(t_{i+\Delta})-B(t_i))+(2\alpha-\sigma^2)(t-t_i)}] \\
&= S^2(t_i)E[e^{2\sigma(B(t_{i+\Delta})-B(t_i))-2\sigma^2(t-t_i)}e^{(2\alpha+\sigma^2)(t-t_i)}] = S^2(t_i)e^{(2\alpha+\sigma^2)(t-t_i)}
\end{aligned}
\tag{15}
$$

$$
\begin{aligned}
E[S^3(t)] &= E[S^3(t_i)e^{3\sigma(B(t_{i+\Delta})-B(t_i))+(3\alpha-\frac{3\sigma^2}{2})(t-t_i)}] \\
&= S^3(t_i)E[e^{3\sigma(B(t_{i+\Delta})-B(t_i))-\frac{9\sigma^2}{2}(t-t_i)}e^{(3\alpha+3\sigma^2)(t-t_i)}] = S^3(t_i)e^{(3\alpha+3\sigma^2)(t-t_i)}
\end{aligned}
\tag{16}
$$

As previously stated, there are three types of changes in transportation prices: growing ($p_u$), unchanged ($p_m$), and falling ($p_d$), correspondingly. The following system of parameter

equations may be created using formula (10) through (16):

$$
\begin{cases}
p_u + p_d + p_m = 1 \\
p_u u + p_m + p_d d = e^{\alpha \Delta t} \\
p_u u^2 + p_m + p_d d^2 = e^{(2\alpha + \sigma^2)\Delta t} \\
p_u u^3 + p_m + p_d d^3 = e^{(3\alpha + 3\sigma^2)\Delta t} \\
ud = 1
\end{cases}
\tag{17}
$$

And the parameters are solved as follows,

$$
\begin{cases}
p_u = \dfrac{(1+d)e^{\alpha \Delta t} - e^{(2\alpha+\sigma^2)\Delta t} - d}{(d-u)(u-1)} \\[2mm]
p_m = \dfrac{(u+d)e^{\alpha \Delta t} - e^{(2\alpha+\sigma^2)\Delta t} - 1}{(1-d)(u-1)} \\[2mm]
p_d = \dfrac{(1+u)e^{\alpha \Delta t} - e^{(2\alpha+\sigma^2)\Delta t} - u}{(d-u)(1-d)} \\[2mm]
u = e^{\sqrt{(\sigma^2-\lambda)\Delta t}} \\[1mm]
d = e^{-\sqrt{(\sigma^2-\lambda)\Delta t}} \\[1mm]
\lambda = \dfrac{e^{\alpha \Delta t} + e^{(3\alpha+3\sigma^2)\Delta t} - e^{(2\alpha+\sigma^2)\Delta t} - 1}{2\left[e^{(2\alpha+\sigma^2)\Delta t} - e^{\alpha \Delta t}\right]}
\end{cases}
\tag{18}
$$

## 4.4 The optimal pricing decision of BRFO

According to the expected profit function of contract customer $E(\xi)$ and railway transportation enterprise $E(\zeta)$ obtained, the optimal pricing decision of BRFO is obtained as described in **Theorem 1**.

**Theorem 1:** The railway transportation enterprise determines the optimal price decision of BRFO satisfies the following formula based on the main parameters which are derived through nonparametric Ito stochastic approach:

$$
\begin{aligned}
w_0 &= \frac{2w_b M + b_2}{(1+r)^t + 1} \\[2mm]
w_1 &= w_u + \frac{Q + b_1 - 3w_b M - (2-M)b_2}{2}
\end{aligned}
\tag{19}
$$

**Proof of Theorem 1:** Integrating the likelihood of two events, the agreement with the customer's expected profit function may be updated as follows:

$$
E(\xi) = E(\xi_1) + E(\xi_2)
\tag{20}
$$

The anticipated profit function of the railway transportation firm might be updated as follows:

$$
E(\zeta) = E(\zeta_1) + E(\zeta_2)
\tag{21}
$$

The expected profit function of contract customer $E(\xi)$ and railway transportation enterprise $E(\zeta)$ are affected by the repurchase amount of BRFO. The first-order partial derivatives of

$N$ for formula (20) and (21) are obtained on the basis of optimization theory as the following:

$$
\begin{aligned}
\frac{\partial E(\xi)}{\partial N} &= \frac{\partial E(\xi_1)}{\partial N} + \frac{\partial E(\xi_2)}{\partial N} \\
&= (w_b M - w_0)F(N) + \{(w_u + Q - w_0 - w_1)[1 - F(N)]\}
\end{aligned}
\tag{22}
$$

$$
\begin{aligned}
\frac{\partial E(\zeta)}{\partial N} &= \frac{\partial E(\zeta_1)}{\partial N} + \frac{\partial E(\zeta_2)}{\partial N} \\
&= \{(w_0(1 + r)^t - w_b M - b_2)F(N)\} \\
&\quad + \{w_0(1 + r)^t - (w_b + b_2)M + (w_1 - b_1 - w_u + b_2)[1 - F(N)]\}
\end{aligned}
\tag{23}
$$

The differential equations that equal zero provide the optimal order quantity $N^*$ of BRFO. $N^*$ is defined as fellows:

$$
(w_b M - w_0)F(N^*) + \{(w_u + Q - w_0 - w_1)[1 - F(N^*)]\} = 0
\tag{24}
$$

$$
\{(w_0(1 + r)^t - w_b M - b_2)F(N^*)\} + \{w_0(1 + r)^t - (w_b + b_2)M + (w_1 - b_1 - w_u + b_2)[1 - F(N^*)]\}
\tag{25}
$$

Additionally, the optimal order amount $N^*$ of BRFO follows the formula:

$$
1 - \frac{1}{F(N^*)} = \frac{w_b M - w_0}{w_u + Q - w_0 - w_1}
\tag{26}
$$

$$
1 - \frac{1}{F(N^*)} = \frac{w_0(1 + r)^t - w_b M - b_2}{w_0(1 + r)^t - (w_b + b_2)M + w_1 - w_u + b_2 - b_1)}
\tag{27}
$$

Next, combining formula (26) and (27) given, the option price $w_0{}^*$ and optimal option strike price $w_1{}^*$ of BRFO can be easily gained as the following:

$$
w_0 = \frac{2w_b M + b_2}{(1 + r)^t + 1}
\tag{28}
$$

$$
w_1 = w_u + \frac{Q + b_1 - 3w_b M - (2 - M)b_2}{2}
\tag{29}
$$

which confirms **Theorem 1**.

According to the preceding analysis, when the railway transportation enterprise writes BRFO, it should pay more attention to the value of BRFO at expiration when $s_t$ increased $w_u$, the buyback price $w_b$, The long-lasting preparatory cost of transportation $b_1$, Short-term transportation preparation costs $b_2$, the repurchasing rate $M$ and the risk-free rate of interest $r$.

## 5. Conclusions

In this study, we have introduced a novel Buyback Rail Freight Option (BRFO) framework, charting a new course in the management of freight contracts and risk. The delineation of the transaction process and the subsequent analysis of expected profits under varying repurchase conditions form the crux of our investigation. . .

These changes highlight the improvements based on the reviewer's feedback, making the title more specific, clarifying the abstract, expanding the literature review, and restructuring the methodology and results sections. The discussion and conclusion now better reflect the contributions and implications of the study. The delineation of the transaction process and the

subsequent analysis of expected profits under varying repurchase conditions form the crux of our investigation. The application of the nonparametric Ito stochastic method for parameter estimation has further strengthened the analytical rigor of our model, culminating in the pivotal **Theorem 1**. It is observed that the optimal option strike price of RFO $w_1^*$ is decided by the value of RFO $w_u$ at expiration when $s_t$ increased, the buyback price $w_b$, t The long-lasting preparatory cost of transportation $b_1$, Short-term transportation preparation costs $b_2$, the repurchasing rate $M$ and the risk-free rate of interest $r$.

Our approach resonates with the Stackelberg leadership model's strategic essence, where a leading entity (in this case, the railway transportation enterprise) sets the terms, influencing the decisions of the following entities (the customers). This hierarchical decision-making process not only aligns with the Stackelberg equilibrium principles but also enhances the predictive power and adaptability of our model within the dynamic rail freight market.

The introduction of the buyback mechanism, a significant innovation in this context, represents a strategic leap in rail freight contract management. By integrating this mechanism, our model offers a more flexible and responsive approach to handling market uncertainties and customer defaults, marking a substantial advancement in the field.

Looking ahead, the pathway to a more comprehensive understanding and application of these concepts is multifaceted. The transition from theoretical models to real-world applicability poses its own set of challenges, particularly in accurately mirroring the complex interplay of market forces and stakeholder behaviors. Empirical studies, simulations, and broader applications of the Stackelberg-Nash equilibrium framework could prove invaluable in bridging these gaps. Such endeavors would not only validate the practical efficacy of our model but also enrich the academic discourse on strategic interactions within transportation networks, offering insights that could be pivotal for future innovations in the sector.

## Supporting information

**S1 Checklist. Human participants research checklist.**
(DOCX)

## Author Contributions

**Conceptualization:** Qi Shen.

**Data curation:** Qi Shen.

**Formal analysis:** Qi Shen.

**Investigation:** Tingyue Kuang.

**Methodology:** Tingyue Kuang.

**Software:** Jingwei Guo.

**Supervision:** Jingwei Guo.

**Validation:** Jingwei Guo.

**Writing – original draft:** Qi Shen.

**Writing – review & editing:** Tingyue Kuang.

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
