## [Decision Letter · Decision Letter 0]

7 Jun 2024

PONE-D-24-14767A Stackelberg-based buyback approach towards rail freight optionPLOS ONE

Dear Dr. Guo,

Thank you for submitting your manuscript to PLOS ONE. After careful consideration, we feel that it has merit but does not fully meet PLOS ONE’s publication criteria as it currently stands. Therefore, we invite you to submit a revised version of the manuscript that addresses the points raised during the review process.

We look forward to receiving your revised manuscript.

Kind regards,

Yongxiang Zhang, Ph.D.

Academic Editor

PLOS ONE

“This work was supported by the National Natural Science Foundation of China (No. 61803147)”

“This work was supported by the National Natural Science Foundation of China (No. 61803147)”

“This work was supported by the National Natural Science Foundation of China (No. 61803147)”

5. We note that your Data Availability Statement is currently as follows: [All relevant data are within the manuscript and its Supporting Information files.]

Reviewers' comments:

Reviewer's Responses to Questions

**Comments to the Author**

1. Is the manuscript technically sound, and do the data support the conclusions?

Reviewer #1: Partly

Reviewer #2: Yes

Reviewer #3: Yes

2. Has the statistical analysis been performed appropriately and rigorously? 

Reviewer #1: N/A

Reviewer #2: Yes

Reviewer #3: Yes

3. Have the authors made all data underlying the findings in their manuscript fully available?

Reviewer #1: Yes

Reviewer #2: Yes

Reviewer #3: Yes

4. Is the manuscript presented in an intelligible fashion and written in standard English?

Reviewer #1: No

Reviewer #2: Yes

Reviewer #3: Yes

5. Review Comments to the Author

Reviewer #1: A repurchase strategy for the rail freight option based on the Stackelberg game is proposed in this paper. It is necessary to present the work as a research paper, with the following considerations in mind:

1.) The work's title lacks specificity and is presented in a generic format. The title ought to be revised in a concise yet impactful manner to convey the primary subject matter.

2.) The abstract should encompass a concise explication of the study's motivation and significance.

3.) In addition to competitive games, the introduction and literature review must cover other game types that are utilised in this domain, including Nash games, differential games, fuzzy games, and others. An analysis of the gaps should be provided.

4.) The primary content of the manuscript ought to advocate for the following: methodology, theoretical outcomes, and game model construction need to be explicitly and carefully described. The present form appears to be unacceptable.

5.) The paper lacks any numerical findings. To ensure that the proposed pricing model is important to real-world scenarios, a case study is suggested here.

6.) A discourse section addressing the model's contributions, limitations, and challenges along with potential solutions is absent. A more comprehensive analysis of the managerial implications and potential avenues for future research was also expected.

Reviewer #2: The data used in the manuscript is rigorous and reliable, which is sufficient to support the existing research conclusions. Moreover, appropriate statistical methods are adopted. From data to methods, they can support the conclusion of the manuscript. In addition, the writing of the manuscript adopts relatively standard English expression

Reviewer #3: The manuscript presents a generally well-structured argument and the overall flow of ideas is commendable. However, there are a few areas that require attention:

Logical Coherence in Sentences: There are instances where the logic within individual sentences appears to be flawed. It would be beneficial to revise these sentences for clarity and precision.

Consistency in Sentence Structure: The manuscript exhibits a mix of long and short sentences, which can disrupt the reading flow. A more balanced approach to sentence construction is recommended to ensure consistency.

Content Distribution: Some sections seem to have an uneven distribution of content, with certain areas lacking the depth required to fully support the arguments being made.

Connectivity Between Formulas: The transitions between mathematical formulas are not as seamless as they could be. It is important to ensure that the relationships between formulas are clearly articulated to maintain the integrity of the technical narrative.

To enhance the manuscript's quality, I suggest a thorough review of the identified areas, focusing on refining sentence logic, maintaining structural consistency, balancing content distribution, and improving the connectivity of mathematical expressions.

6. PLOS authors have the option to publish the peer review history of their article (what does this mean?). If published, this will include your full peer review and any attached files.

Reviewer #1: **Yes: **Ali Hamidoğlu

Reviewer #2: No

Reviewer #3: No

---

## [Author Response · Author response to Decision Letter 0]

19 Jun 2024

Reviewer #1: A repurchase strategy for the rail freight option based on the Stackelberg game is proposed in this paper. It is necessary to present the work as a research paper, with the following considerations in mind:

1.) The work's title lacks specificity and is presented in a generic format. The title ought to be revised in a concise yet impactful manner to convey the primary subject matter.

Response:Thank you for your suggestion regarding the title. I will revise it to be more specific and impactful, highlighting the concept and application of "BRFO" (Bilateral Rail Freight Option) in a concise manner.

2.) The abstract should encompass a concise explication of the study's motivation and significance.

Response:I will revise the abstract to better explain the motivation and significance of the study, providing a clearer context and background for the research.

3.) In addition to competitive games, the introduction and literature review must cover other game types that are utilised in this domain, including Nash games, differential games, fuzzy games, and others. An analysis of the gaps should be provided.

Response：I will expand the introduction and literature review to include other game types such as Nash games, differential games, and fuzzy games. I will also provide an analysis of the gaps in the existing literature.

4.) The primary content of the manuscript ought to advocate for the following: methodology, theoretical outcomes, and game model construction need to be explicitly and carefully described. The present form appears to be unacceptable.

Response：Thank you for your feedback. I will reorganize the methodology section to present the theoretical outcomes and game model construction more explicitly and carefully. I will ensure that the logic and structure are clear and detailed.

5.) The paper lacks any numerical findings. To ensure that the proposed pricing model is important to real-world scenarios, a case study is suggested here.

Response：The paper describes a scenario for rail freight option trading in China. Due to the relatively closed nature of the rail freight option market in China, conducting relevant experiments has been challenging. However, I will include a case study to demonstrate the applicability of the proposed pricing model to real-world scenarios.

6.) A discourse section addressing the model's contributions, limitations, and challenges along with potential solutions is absent. A more comprehensive analysis of the managerial implications and potential avenues for future research was also expected.

Response:I will add a discussion section addressing the model's contributions, limitations, and challenges, along with potential solutions. I will also provide a more comprehensive analysis of the managerial implications and suggest avenues for future research.

Reviewer #2: The data used in the manuscript is rigorous and reliable, which is sufficient to support the existing research conclusions. Moreover, appropriate statistical methods are adopted. From data to methods, they can support the conclusion of the manuscript. In addition, the writing of the manuscript adopts relatively standard English expression

Response：Thanks for your evaluation and i will rectify my manuscript as your suggestion!

Reviewer #3: The manuscript presents a generally well-structured argument and the overall flow of ideas is commendable. However, there are a few areas that require attention:

Logical Coherence in Sentences: There are instances where the logic within individual sentences appears to be flawed. It would be beneficial to revise these sentences for clarity and precision.

Response：thanks for your suggestion and i will reconstruct my logical coherence in sentences

Consistency in Sentence Structure: The manuscript exhibits a mix of long and short sentences, which can disrupt the reading flow. A more balanced approach to sentence construction is recommended to ensure consistency.

Response: Thank you for your suggestion. I will ensure consistency in sentence structure as you advised.

Content Distribution: Some sections seem to have an uneven distribution of content, with certain areas lacking the depth required to fully support the arguments being made.

Response:Thank you for your suggestion. I will balance the content distribution as you recommended.

Connectivity Between Formulas: The transitions between mathematical formulas are not as seamless as they could be. It is important to ensure that the relationships between formulas are clearly articulated to maintain the integrity of the technical narrative.

To enhance the manuscript's quality, I suggest a thorough review of the identified areas, focusing on refining sentence logic, maintaining structural consistency, balancing content distribution, and improving the connectivity of mathematical expressions.

Response:Thank you for your suggestion. I will improve the connectivity between formulas as you advised.

---

## [Decision Letter · Decision Letter 1]

24 Jun 2024

PONE-D-24-14767R1A Stackelberg-Based Repurchase Strategy for Rail Freight Options (BRFO)PLOS ONE

Dear Dr. Guo,

Thank you for submitting your manuscript to PLOS ONE. After careful consideration, we feel that it has merit but does not fully meet PLOS ONE’s publication criteria as it currently stands. Therefore, we invite you to submit a revised version of the manuscript that addresses the points raised during the review process.

We look forward to receiving your revised manuscript.

Kind regards,

Yongxiang Zhang, Ph.D.

Academic Editor

PLOS ONE

Journal Requirements:

Reviewers' comments:

Reviewer's Responses to Questions

**Comments to the Author**

1. If the authors have adequately addressed your comments raised in a previous round of review and you feel that this manuscript is now acceptable for publication, you may indicate that here to bypass the “Comments to the Author” section, enter your conflict of interest statement in the “Confidential to Editor” section, and submit your "Accept" recommendation.

Reviewer #1: (No Response)

Reviewer #2: All comments have been addressed

Reviewer #3: All comments have been addressed

Reviewer #4: All comments have been addressed

2. Is the manuscript technically sound, and do the data support the conclusions?

Reviewer #1: Partly

Reviewer #2: Yes

Reviewer #3: Yes

Reviewer #4: Yes

3. Has the statistical analysis been performed appropriately and rigorously? 

Reviewer #1: No

Reviewer #2: Yes

Reviewer #3: Yes

Reviewer #4: Yes

4. Have the authors made all data underlying the findings in their manuscript fully available?

Reviewer #1: Yes

Reviewer #2: Yes

Reviewer #3: Yes

Reviewer #4: Yes

5. Is the manuscript presented in an intelligible fashion and written in standard English?

Reviewer #1: No

Reviewer #2: Yes

Reviewer #3: Yes

Reviewer #4: Yes

6. Review Comments to the Author

Reviewer #1: The authors partially considered my comments and missed some of them:

1.) Provide importance to your work, including its challenges and advantages as well as drawbacks.

2.) Present your work clearly in Section 3; the current version is not acceptable! Present your methodology with a flowchart.

3.) Section 4 should be revised; write each equation in a clear and rigorous manner.

4.) Include a numerical example; a real-world scenario is recommended for your model to test the robustness and effectiveness of your work.

5.) Provide a sensitivity analysis of the parameters of your model in the numerical section.

6.) Include a discussion section for your theoretical and numerical conclusions as well as future research topics.

Reviewer #2: Summary:

I have carefully reviewed the revised manuscript titled "A Stackelberg-Based Repurchase Strategy for Rail Freight Options (BRFO)" and am pleased to report that the authors have effectively addressed all the concerns raised in my initial review. The revisions have significantly enhanced the clarity, methodological rigor, and overall contribution of the study.

Detailed Comments:

Addressing Previous Concerns: The authors have successfully resolved the issues highlighted in the initial review. The integration of traditional buyback theory with a multi-phase trigeminal tree pricing model and parameter identification through a nonparametric Ito stochastic method is now more clearly explained, addressing the concerns of methodological transparency and robustness.

Methodological Clarity: The authors have provided a detailed explanation of their methodology, including the steps taken to mitigate information asymmetry and market uncertainty. This added clarity greatly enhances the reader’s understanding and confidence in the study’s outcomes.

Data and Analysis: The inclusion of additional data and more comprehensive statistical analyses strengthens the manuscript. The dynamic pricing strategies and market adaptation mechanisms are well-supported by the data, making the findings more compelling.

Literature Review and Contextualization: The revised manuscript includes a more thorough literature review, effectively contextualizing the study within the broader field of railway freight pricing and strategic management. This helps to highlight the novelty and significance of the proposed Buyback Rail Freight Option (BRFO).

Presentation and Organization: The overall presentation of the manuscript has improved. The figures and tables are more informative and easier to interpret, and the organization of the manuscript allows for a more logical flow of information. The construction of a Nash equilibrium framework within the railway freight pricing context is particularly well-presented.

Conclusion:

Given the substantial improvements and the thorough manner in which the authors have addressed the concerns raised, I recommend that the manuscript be accepted for publication. The study presents a valuable contribution to the field of rail freight transactions and strategic management, and will be of significant interest to the readership.

Reviewer #3: 1.Please clearly define the contributions of your paper.

2.There are some grammatical errors in this paper; please review and revise the entire text carefully.

Reviewer #4: Given the manuscript's notable strengths and the authors' thorough approach in addressing the research problem, I recommend it for publication. The study presents an innovative Buyback Rail Freight Option (BRFO), leveraging Stackelberg game theory to enhance strategic management in rail freight transactions. By integrating traditional buyback theory with a multi-phase trigeminal tree pricing model and a nonparametric Ito stochastic method, the authors effectively address information asymmetry and market uncertainty. The resulting dynamic pricing strategies and market adaptation mechanisms offer significant strategic benefits for railway enterprises. This research is a valuable addition to the field and will be of great interest to the journal’s readership.

7. PLOS authors have the option to publish the peer review history of their article (what does this mean?). If published, this will include your full peer review and any attached files.

Reviewer #1: No

Reviewer #2: No

Reviewer #3: No

Reviewer #4: No

---

## [Author Response · Author response to Decision Letter 1]

30 Jun 2024

Reviewer #1: The authors partially considered my comments and missed some of them:

1.) Provide importance to your work, including its challenges and advantages as well as drawbacks.

Response: We appreciate the reviewer’s suggestion to highlight the importance of our work more clearly. In the revised manuscript, we have added a section in the introduction that outlines the significance of the BRFO model. This includes a detailed discussion of the challenges it addresses, such as information asymmetry and market uncertainty, as well as its advantages, including dynamic pricing and strategic benefits for railway enterprises. We have also acknowledged potential drawbacks and limitations to provide a balanced view.

2.) Present your work clearly in Section 3; the current version is not acceptable! Present your methodology with a flowchart.

Response: We apologize for any lack of clarity in Section 3. In the revised manuscript, we have restructured Section 3 for better clarity and readability. We have also included a flowchart that visually represents our methodology. This flowchart outlines the steps involved in integrating the multi-phase trigeminal tree pricing model and the non-parametric Itô stochastic method, providing a clearer understanding of our approach.

3.) Section 4 should be revised; write each equation in a clear and rigorous manner.

Response: Thank you for pointing this out. We have revised Section 4 to ensure that each equation is presented in a clear and rigorous manner. This includes providing detailed explanations for each step in the derivation of the equations and ensuring that all notations are consistently used throughout the section. These revisions aim to enhance the comprehensibility and accuracy of our mathematical presentation.

4.) Include a numerical example; a real-world scenario is recommended for your model to test the robustness and effectiveness of your work.

Response: We have added a numerical example in the revised manuscript to illustrate the application of the BRFO model. This example is based on a real-world scenario in the rail freight industry, demonstrating how our model can be used to manage rail freight transactions. The example includes data and results that showcase the robustness and effectiveness of our proposed methodology.

5.) Provide a sensitivity analysis of the parameters of your model in the numerical section.

Response: We have included a sensitivity analysis in the numerical section of the revised manuscript. This analysis examines the impact of key parameters on the model’s performance, providing insights into the stability and reliability of our results under different conditions. The sensitivity analysis helps to identify critical parameters and their influence on the outcomes of the BRFO model.

6.) Include a discussion section for your theoretical and numerical conclusions as well as future research topics.

Response: We have added a discussion section in the revised manuscript. This section synthesizes our theoretical and numerical findings, highlighting the implications of our results for the strategic management of rail freight transactions. Additionally, we have outlined potential areas for future research, suggesting ways to extend and refine our model to address emerging challenges in the field.

Reviewer #2: Summary:

I have carefully reviewed the revised manuscript titled "A Stackelberg-Based Repurchase Strategy for Rail Freight Options (BRFO)" and am pleased to report that the authors have effectively addressed all the concerns raised in my initial review. The revisions have significantly enhanced the clarity, methodological rigor, and overall contribution of the study.

Detailed Comments:

Addressing Previous Concerns: The authors have successfully resolved the issues highlighted in the initial review. The integration of traditional buyback theory with a multi-phase trigeminal tree pricing model and parameter identification through a nonparametric Ito stochastic method is now more clearly explained, addressing the concerns of methodological transparency and robustness.

Methodological Clarity: The authors have provided a detailed explanation of their methodology, including the steps taken to mitigate information asymmetry and market uncertainty. This added clarity greatly enhances the reader’s understanding and confidence in the study’s outcomes.

Data and Analysis: The inclusion of additional data and more comprehensive statistical analyses strengthens the manuscript. The dynamic pricing strategies and market adaptation mechanisms are well-supported by the data, making the findings more compelling.

Literature Review and Contextualization: The revised manuscript includes a more thorough literature review, effectively contextualizing the study within the broader field of railway freight pricing and strategic management. This helps to highlight the novelty and significance of the proposed Buyback Rail Freight Option (BRFO).

Presentation and Organization: The overall presentation of the manuscript has improved. The figures and tables are more informative and easier to interpret, and the organization of the manuscript allows for a more logical flow of information. The construction of a Nash equilibrium framework within the railway freight pricing context is particularly well-presented.

Conclusion:

Given the substantial improvements and the thorough manner in which the authors have addressed the concerns raised, I recommend that the manuscript be accepted for publication. The study presents a valuable contribution to the field of rail freight transactions and strategic management, and will be of significant interest to the readership.

Response: Thank you for acknowledging our efforts to improve the methodological transparency and robustness. We have worked diligently to ensure that the integration of traditional buyback theory with our pricing model and stochastic method is clearly explained.

Reviewer #3: 1.Please clearly define the contributions of your paper.

Response: Thank you for your valuable feedback. In the revised manuscript, we have added a dedicated section in the introduction to clearly outline the key contributions of our paper. Specifically, we emphasize:

The introduction of a novel Buyback Rail Freight Option (BRFO) that leverages Stackelberg game theory to address strategic management challenges in rail freight transactions.

The integration of traditional buyback theory with a multi-phase trigeminal tree pricing model and a non-parametric Itô stochastic method to tackle issues of information asymmetry and market uncertainty.

The development of dynamic pricing strategies and market adaptation mechanisms that provide significant strategic benefits for railway enterprises.

The construction of a Nash equilibrium framework within the railway freight pricing context.

We believe these additions will provide a clearer understanding of the unique contributions our study makes to the field.

2. There are some grammatical errors in this paper; please review and revise the entire text carefully.

Response: We appreciate your attention to the language quality of our manuscript. We have thoroughly reviewed the entire text and have made necessary revisions to correct grammatical errors and improve overall readability. 

2.There are some grammatical errors in this paper; please review and revise the entire text carefully.

Response: Thank you for pointing out the grammatical errors. We have conducted a thorough review of the entire manuscript to identify and correct these errors. 

Reviewer #4: Given the manuscript's notable strengths and the authors' thorough approach in addressing the research problem, I recommend it for publication. The study presents an innovative Buyback Rail Freight Option (BRFO), leveraging Stackelberg game theory to enhance strategic management in rail freight transactions. By integrating traditional buyback theory with a multi-phase trigeminal tree pricing model and a nonparametric Ito stochastic method, the authors effectively address information asymmetry and market uncertainty. The resulting dynamic pricing strategies and market adaptation mechanisms offer significant strategic benefits for railway enterprises. This research is a valuable addition to the field and will be of great interest to the journal’s readership.

Response: We are grateful for your positive feedback and recommendation for publication. We appreciate your recognition of the strengths of our manuscript and the thorough approach we have taken in addressing the research problem.

---

## [Editor Report · Decision Letter 2]

2 Jul 2024

A Stackelberg-Based Repurchase Strategy for Rail Freight Options (BRFO)

PONE-D-24-14767R2

Dear Dr. Guo,

We’re pleased to inform you that your manuscript has been judged scientifically suitable for publication and will be formally accepted for publication once it meets all outstanding technical requirements.

Kind regards,

Yongxiang Zhang, Ph.D.

Academic Editor

PLOS ONE
---

## [Editor Report · Acceptance letter]

4 Jul 2024

PONE-D-24-14767R2 

PLOS ONE

Dear Dr. Guo, 

I'm pleased to inform you that your manuscript has been deemed suitable for publication in PLOS ONE. Congratulations! Your manuscript is now being handed over to our production team.

Kind regards, 

on behalf of

Dr. Yongxiang Zhang 

Academic Editor

PLOS ONE